# Flat-Band in Pyrochlore Oxides: A First-Principles Study

**DOI:** 10.3390/nano9060876

**Published:** 2019-06-10

**Authors:** Izumi Hase, Takashi Yanagisawa, Kenji Kawashima

**Affiliations:** 1National Institute of Advanced Industrial Science and Technology (AIST), Tsukuba 305-8568, Japan; t-yanagisawa@aist.go.jp; 2IMRA Material R&D Co. Ltd., Kariya 448-0032, Japan; kenji.kawashima@aisin.co.jp

**Keywords:** pyrochlore oxide, band calculation, flat band, ferromagnetism

## Abstract

Using a first-principles electronic band calculation, we obtained a quasi flat-band near the Fermi level for the six pyrochlore oxides, A_2_B_2_O_7_. These quasi flat-bands are mostly characterized by the s-orbitals of the A-site. The band structures of these oxides are well described by the non-interacting Mielke model. Spin-polarized calculations showed that the ground state of these compounds was ferromagnetic after appropriate carrier doping, despite the absence of the magnetic element.

## 1. Introduction

Magnetic oxides have been studied at length, and are of great importance from both the scientific and applied standpoint. They include Mn-perovskites, hexaferrites [1,2,3,4,5,6,7,8,9,10], and so on. Among them, pyrochlore oxides denoted by A_2_B_2_O_7_ (or A_2_B_2_O_6_O’) have been given much attention due to their rich variety of physical properties [11,12]. Most of the 150 or more kinds of pyrochlore oxides have been explored, and most have open-shell d-orbitals and f-orbitals on the A-site or the B-site. Usually, these orbitals are localized and only have spin degree of freedom. Since both of the A-sublattice and B-sublattice in A_2_B_2_O_7_ are geometrically frustrated, these localized spins on the frustrated sublattice (i.e., pyrochlore lattice) bring about many exotic magnetic properties such as spin-ice, quantum spin liquid, and even a “magnetic monopole” [13,14,15,16,17]. There is another type of pyrochlore oxide, Cd_2_Re_2_O_7_, that shows superconductivity [18,19]. In Cd_2_Re_2_O_7_, the d-orbitals form itinerant bands, contrary to the above-mentioned localized d- (or f-) orbital system. In this paper, we studied the third type of pyrochlore oxides, which do not include relevant d- or f- orbitals. In particular, we focused on compounds with the relevant bands composed of s-orbitals of the A-site. At first glance, this type of pyrochlore oxide seems to be quite trivial due to the absence of strong correlation on the d- or f- orbitals, but in fact, may show many interesting properties. When we only considered one s-orbital on each site of the pyrochlore lattice, and only considered the nearest-neighbor transfer (*t*), it gave the (non-interacting) Mielke model described by the following Hamiltonian:(1)$$H_{0}=-t\underset{<i,j>\sigma }{\sum }c_{i\sigma }^{†}c_{j\sigma }$$
where $c_{i\sigma }$ denotes the annihilation operator at *i*-th site with spin *σ*, and <*i*,*j*> denotes nearest neighbors. We can directly diagonalize this Hamiltonian in momentum space [20] and obtain the energy eigenvalues as
(2)$$E_{1,2}=-2t\left[1\pm \sqrt{1+A_{k}}\right],\;E_{3,4}=2t$$
where $A_{k}=\cos\left(2k_{x}\right)\cos\left(2k_{y}\right)+\cos\left(2k_{y}\right)\cos\left(2k_{z}\right)+\cos\left(2k_{z}\right)\cos\left(2k_{x}\right)$. The energy bands *E*_3,4_ do not depend on the wave vector ***k*** and are called flat-bands (FBs). Since the density of states (DOS) of FB is singular, this model gives many anomalous physical properties. In fact, FBs are seen in many lattice models [21,22], and vigorous studies have predicted that many attractive physical properties such as exact ferromagnetism [23,24], high-temperature superconductivity [25], various topological states [20,26], and the fractional quantum Hall effect [27] will be developed. Therefore, it is very important to find real materials that are approximately described by the FB model. Figure 1 shows the correspondence between the localized spin system and the itinerant system. 

In our previous study [28], we proposed a guiding principle to obtain FB near the Fermi level (*E*_F_). The point was to use “s2” ions, in other words, the ions that had two electrons on the outermost s-orbitals. For example, Sn^2+^, Tl^1+^, and Bi^3+^ are all s2 ions. The recipe is very simple: (1) Put the s2 ions on the pyrochlore A-site; (2) Fill other sites by closed-shell ions by keeping charge neutrality (e.g., Sn^2+^_2_Nb^5+^_2_O^2−^_7_); and (3) Calculate the band structure and check whether FB is found or not. There are many candidates with FB chosen by steps (1) and (2), but we have shown that only the (+2, +5) type, i.e., A^2+^_2_B^5+^_2_O^2−^_7_ may exhibit clear (i.e., without being masked by other bands) FB [28]. Due to the large DOS at the top of the valence band, these compounds may show a ferromagnetic ground state when some holes are doped [28,29]. 

Here, we considered the situation of the “s1” counterpart. Namely, the relevant outermost s-orbital of the A-site is half-filled, in other words, occupied by one electron per site. For example; Tl^2+^, Pb^3+^, and Bi^4+^ are all s1 ions. This situation has been discussed in the context of the valence skipper [30,31,32]. The high-temperature superconductivity of Ba(Pb_1−x_Bi_x_)O_3_ and (Ba_1−x_K_x_)BiO_3_ has been explained by the valence-skipping mechanism, accompanied with the possible charge order of s0 (Bi^5+^) and s2 (Bi^3+^) [33,34,35,36,37]. Generally speaking, the s1 state is chemically unstable and splits into s0 and s2 states. In other words, two electrons naturally form a pair and move around. This process effectively generates attractive interaction *U* < 0. Actually, the negative-*U* Hubbard model on a bipartite lattice can lead to s-wave superconductivity [38,39]. However, the negative-U Hubbard model on a frustrated lattice has been much less investigated. In this sense, the s1 compounds Tl_2_Nb_2_O_7_ and Tl_2_Ta_2_O_7_ provide good examples, which may realize this interesting situation [40,41,42]. Unfortunately, they are yet to succeed in metalizing or superconducting Tl_2_Nb_2_O_7_. Nevertheless, this direction of research may foresee the manifestation of rich physical properties as above-mentioned. In relation to the s2 system, if the underlying band structure is similar, then the s1 system may have a ferromagnetic ground state when the electrons are doped.

In this paper, we demonstrate that the valence bands of six pyrochlore oxides with A_2_B_2_O_7_ (A = Sn, Pb, Tl; B = Nb, Ta) can be approximately described by the above-mentioned Mielke model. Moreover, we theoretically show that appropriate carrier doping brings about a ferromagnetic ground state, even without magnetic elements. In Section 2, we describe the method of calculation, and in Section 3 we present the results and discussion. 

## 2. Computational Methods 

We calculated six pyrochlore oxides with A_2_B_2_O_7_ (A = Sn, Pb, Tl; B = Nb, Ta) from first-principles and used the density-functional theory (DFT) and a full-potential linearized augmented plane wave (FLAPW) scheme (WIEN2k code [43]). The exchange-correlation potential was constructed within the generalized gradient approximation [44]. In Section 3.2, the results are shown for the exchange-correlation potential with local density approximation (LDA) [45]. The parameter *RK*_max_ was chosen to be 7.0. The *k*-point mesh was taken so that the total number of the mesh in the first Brillouin zone was about 1000. For simplicity, we assumed that they all had an ideal A_2_B_2_O_6_O’ pyrochlore structure with the space group Fd-3m (#227). Since oxygen atoms occupy two crystallographic sites, we named them O and O’ to distinguish between them. The atomic position was A(0,0,0), B(1/2,1/2,1/2), O(*u*,1/8,1/8), and O’(1/8,1/8,1/8). We optimized parameter *u* by minimizing the Hellmann–Feynman force, and the convergence of the atomic position was judged by the Hellmann–Feynman force working on each atom, which was less than 1.0 mRy/a.u. Regarding Tl_2_Ta_2_O_7_, we also optimized the lattice constant *a*. The optimized value *a* = 10.716 Å agreed well with an experimental value *a* = 10.56 Å [46] or 10.651 Å [47]. For the other five compounds, we used the experimental lattice constant. 

Since these compounds have a quasi-FB near *E*_F_, we also performed spin-polarized calculations for doped compounds. For the s2 (s1) compounds, we required hole (electron) doping in order to obtain the magnetic ground state. We chose A_2_B_2_O_6_N for s2 compounds (A = Sn, Pb), and A_2_B_2_O_6_F for s1 compounds (A = Tl).

## 3. Results and Discussion

### 3.1. Band Structure and Quasi Flat Band 

Figure 2 shows the band structure of A_2_Nb_2_O_7_ (A = Pb and Tl). These looked similar to each other, and our previous results for Sn_2_Nb_2_O_7_ also showed a similar band structure [29]. This shows that their electronic structure is described by a unified manner. A quasi-FB was found at the top of the valence band in Sn_2_Nb_2_O_7_ and Pb_2_Nb_2_O_7_, while its bandwidth was different (~0.4 eV for Sn_2_Nb_2_O_7_ and ~0.2 eV for Pb_2_Nb_2_O_7_). Since Tl_2_Nb_2_O_7_ has one less electron per formula unit than Pb_2_Nb_2_O_7_, this quasi-FB becomes unoccupied and *E*_F_ lies just below this quasi-FB. The left panel shows the band structure of the Mielke model. Although this model includes only one adjustable parameter (*t*), its agreement with the band structure of the real compound Pb_2_Nb_2_O_7_ was quite impressive. In the case of the Mielke model shown in Figure 2a, the FB was doubly degenerated. In real A_2_Nb_2_O_7_, the degeneracy is partly lifted, but this energy splitting is almost negligible. The density of states analysis showed that this quasi-FB was mainly composed of A-s and O’-p orbitals. The Nb-d orbitals formed a conduction band and were not relevant for this study. 

The band structure of A_2_Ta_2_O_7_ was essentially similar to its Nb counterpart except for the magnitude of the band gap between the A-s band and Nb/Ta-d band. Since the Ta-d orbital was higher than Nb-d in energy, the band gap of A_2_Ta_2_O_7_ was 0.5–0.7 eV larger than its Nb counterpart. 

### 3.2. Ferromagnetic State

It is known that the Mielke model of Equation (1) has a unique ferromagnetic ground state if any non-vanishing value of the on-site Coulomb interaction (Hubbard *U*) is imposed when the FB is half-filled [23,24]. In the case of a more realistic model including the next nearest neighbor transfer *t*’, the exact solution is unknown. However, there has been a numerical study for the 2D checkerboard lattice model that also has a FB when merely *t* is considered [47]. This shows that the ferromagnetic ground state is still stable when the width of the quasi-FB is smaller than *U*. This result gives an interesting possibility that even a small *U* in a “nonmagnetic” element such as Sn and O can induce ferromagnetism. We performed a spin-polarized first-principles calculation for the hole-doped s2 compounds A_2_B_2_O_6_N (A = Sn, Pb) and the electron-doped s1 compounds A_2_B_2_O_6_F (A = Tl).

Table 1 shows the calculated magnetic moment of A_2_B_2_O_6_X (X = N, F) per primitive unit cell. Since the primitive unit cell of the pyrochlore structure contains four A atoms, the relevant part of the valence band can be described by the Mielke model including four A-s orbitals. All of the compounds in Table 1 have two holes in the valence band, which means that the doubly degenerated quasi-FB is half-filled. Therefore, if the bandwidth of the quasi-FB is smaller than the exchange splitting, it gives the perfect spin polarization with the magnetic moment *M* = 2.0 μ_B_. From Table 1, we can see that this condition is satisfied for hole-doped s2 compounds. On the other hand, hole-doped Tl (s1) compounds have a smaller magnetic moment. This is partly due to the smaller exchange splitting in the Tl system, and partly due to the larger modification of the quasi-FB with respect to the anion substitution in the Tl system.

To see this situation more clearly, we present the band structure of Pb_2_Ta_2_O_6_N and Tl_2_Nb_2_O_6_F in Figure 3. In Pb_2_Ta_2_O_6_N, the effect of substituting N for O was not especially large. The shape of the valence band was mostly retained, however, in Tl_2_Nb_2_O_6_F, replacing O with F resulted in a very large change in the band structure. This may be explained as follows: N is more covalent than O and F, and will be closer to N^2−^ than N^3−^. On the other hand, O and F are mostly ionic and can be written as O^2−^ and F^−^. As a result, the electrostatic potential felt on the A site may not be so different when O is substituted by N, but will be significantly different when O is substituted by F. This “rigid-band” situation is also observed in a system where Sn_2_Ta_2_O_7_ is doped with N [29].

It is known that the existence of ferromagnetism is especially sensitive to the choice of the approximation for the exchange-correlation potential, and to the value of the lattice constant *a* [48,49]. Therefore, we further performed LDA calculations and compared them with the GGA results for Pb_2_Ta_2_O_6_N and Tl_2_Ta_2_O_6_F. We also changed *a* and calculated *M* for each *a* where the results are shown in Figure 4. As for Pb_2_Ta_2_O_6_N, we can see that LDA and GGA had almost the same *M* for each *a*. We found that *M* = 2.00 for all *a*, which shows that *M* was “saturated”, i.e., the up-spin FB was completely filled and the down-spin FB was empty. On the other hand, in Tl_2_Ta_2_O_6_F, both the up-spin band and the down-spin band were partially filled so the magnetic moment had a fractional value. *M*(LDA) is smaller than *M*(GGA), which agrees with the general trend for itinerant ferromagnetism [48,49]. The equilibrium value of *a* for LDA was smaller than that for GGA, which also agrees with the general trend. We could see that the *a* dependence of *M(a)* was almost negligible for all of these cases. In other words, the obtained magnetic moment was almost constant by applying pressure at least up to ~10 GPa. 

### 3.3. Comparison with Experimental Results

We also predicted the ferromagnetic ground state for hole (electron) doped s2 (s1) pyrochlore oxides A_2_B_2_O_7_. Experimentally, small amounts of holes can be doped into Sn_2_Nb_2_O_7_ and Sn_2_Ta_2_O_7_ [50]. However, the ferromagnetic signal has not been reported in the hole doped sample yet. Photoemission spectra of this sample did not show the sharp peak predicted by the expected FB. Møssbauer spectroscopy revealed that this sample had a large amount of Sn deficiency [51]. This deficiency may greatly affect FB, and it is expected that a less Sn-deficient sample may cause unusual magnetic properties. As for Pb_2_Nb_2_O_7_, there was controversy as to whether it was ferroelectric or not, and many crystal structures with lower symmetry have been proposed [52]. However, the lattice instability in the high-symmetry phase mostly occurs in the insulating phase, and an appropriate carrier doping (if possible) may restore the high-symmetry cubic pyrochlore structure. As for Tl_2_Nb_2_O_6+*x*_, the crystal structure is retained as a cubic pyrochlore structure with varying oxygen content *x*. However, metallic samples have not been obtained yet [40,41]. Introducing sufficient numbers of itinerant carriers is a crucial key to realizing this unusual magnetism, which does not essentially include a magnetic element.

When we compare the theory and the experiment, we must take into account the many factors that are not included in the theory. Stoichiometry and oxygen deficiency are also crucial parameters for electronic and magnetic properties in “conventional” magnetic oxides [5,6,7,8,9,10]. The finite size effect due to the sample surface and grain boundary also needs to be considered. All these points are future tasks. 

## 4. Conclusions

We found six pyrochlore oxides A_2_B_2_O_7_ with a quasi flat-band at the top of the valence band. The band dispersion was approximately described by the non-interacting Mielke model. We predict that appropriate carrier doping may induce a ferromagnetic ground state, though they do not contain a magnetic element. Essentially, this ferromagnetic state comes from the high density of states of the quasi flat-band.

In order to realize this “d- and f- orbital free” novel magnetism, we must proceed from both theoretical and experimental points of view. Theoretically, there is the desire to go beyond DFT by including the many-body effect. The calculation for the finite temperature is also needed. How to handle randomness such as the oxygen deficiency that cannot be avoided in actual samples is also a future subject of research. In “conventional” localized d-electron magnetic oxides, randomness often induces the spin-glass behavior [5,6,7,8,9,10]. However, we do not know whether or not spin-glass behavior occurs in the flat band system. Regarding ferromagnetism, the smaller the randomness, the better. Therefore, it will be an experimental challenge to synthesize a crystal with good quality. 

## Figures and Tables

**Figure 1 nanomaterials-09-00876-f001:**
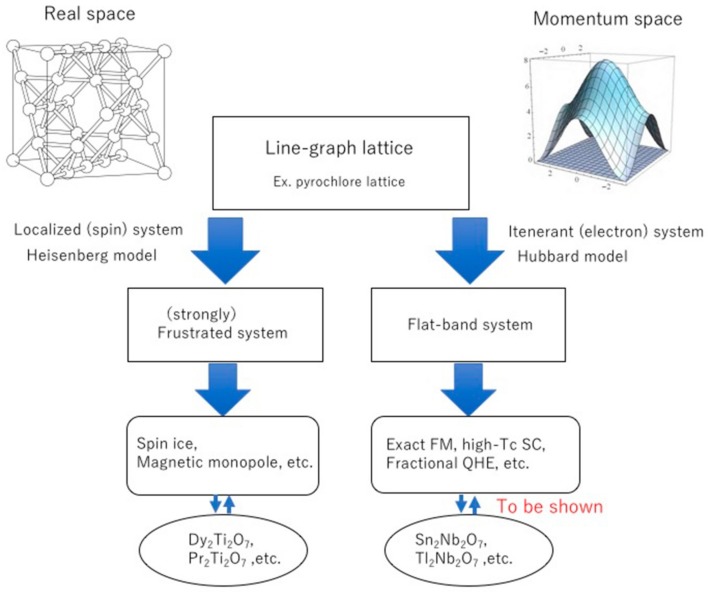
Schematic correspondence between the localized system and the itinerant system.

**Figure 2 nanomaterials-09-00876-f002:**
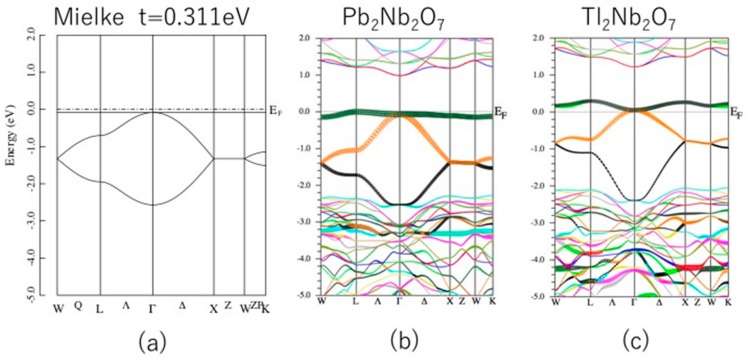
(**a**) Energy band of the Mielke model with *t* = 0.311 eV. (**b**) Band structure of Pb_2_Nb_2_O_7_ and (**c**) Tl_2_Nb_2_O_7_. The spin-orbit interaction is not included. The unit of the vertical axis is eV.

**Figure 3 nanomaterials-09-00876-f003:**
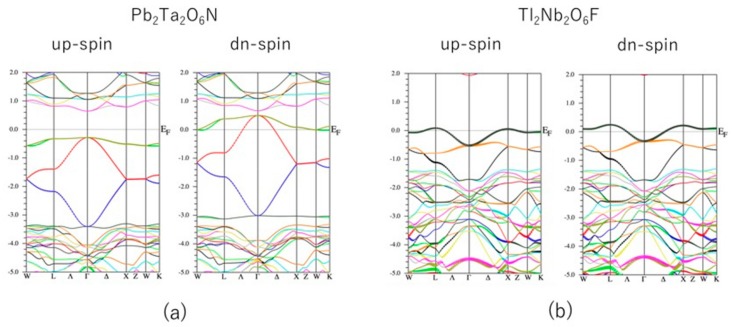
Spin-dependent band structure of (**a**) Pb_2_Ta_2_O_6_N and (**b**) Tl_2_Nb_2_O_6_F. The unit of the vertical axis is eV.

**Figure 4 nanomaterials-09-00876-f004:**
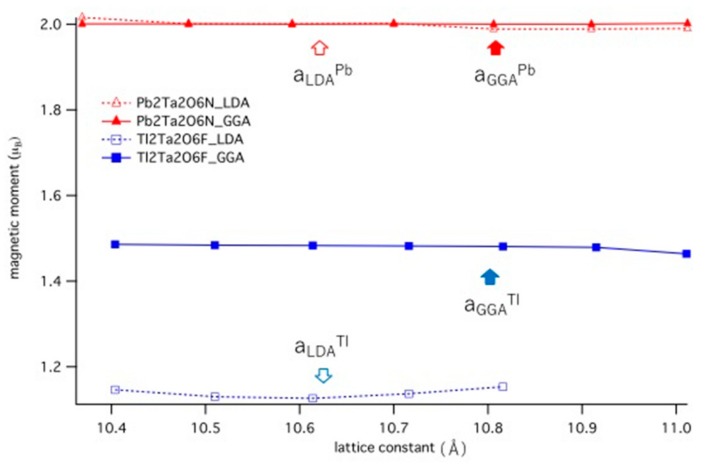
Calculated magnetic moment of Pb_2_Ta_2_O_6_N (triangles) and Tl_2_Ta_2_O_6_F (squares). Open markers are for LDA, and filled markers are for GGA. The arrows show the equilibrium lattice constant.

**Table 1 nanomaterials-09-00876-t001:** Magnetic moment per primitive unit cell (i.e., two formula units) obtained by our first-principles calculation. We show the results for GGA. The lattice constant *a* was set to the value of the “mother compound”. For example, we used the *a* of Sn_2_Nb_2_O_7_ for Sn_2_Nb_2_O_6_N.

Compound	Magnetic Moment (μ_B_ cell)
Sn_2_Nb_2_O_6_N	2.000
Sn_2_Ta_2_O_6_N	2.001
Pb_2_Nb_2_O_6_N	2.001
Pb_2_Ta_2_O_6_N	2.000
Tl_2_Nb_2_O_6_F	1.522
Tl_2_Ta_2_O_6_F	1.507

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
