# Peer review of "Flat-Band in Pyrochlore Oxides: A First-Principles Study"

_nanomaterials, 2019, doi:10.3390/nano9060876_

Reviewer 1 Report

The paper brings new results on electronic structure of six pyrochlore structure oxides A2B2O7 (A = Sn, Pb, Tl; B = Nb, Ta).
It is interesting that these compounds may be ferromagnetic when doped by N or F.  The paper is written concisely and comprehensively and
the results are sufficiently novel and of broad interest to warrant the publication in Nanomaterials.

Regrettably, it seems to me that the findings obtained are not very reliable, as the calculations were performed under very limiting assumptions (within the GGA only, which enhances ferromagnetism, and in 5 oxides the lattice constants were not optimized). I summarize my criticism and hints for improvement below.

1. p. 3, lines 90-91:The authors write:
"As for other five compounds, we used the experimental lattice constant." Why? Why the authors have not optimized the lattice constant also for the other five oxides studied in their paper?
2. p. 5, lines 163-165: The authors write:
"In theoretical point of view, it must be noted that this DFT calculation tends to overestimate the ferromagnetic ground state. Although DFT itself is rigorous for ground state (i.e. zero temperature), exchange-correlation potential is merely an approximated one."
These statements are misleading. DFT is always rigorous for the ground state, but, in practice, in all calculations for solid state I know the exchange-correlation potential is  a l w a y s  approximated, as we do not know its exact form. Further, it is true that the GGA DFT calculations tend to overestimate the ferromagnetic state. On the other hand, the LDA calculations rather suppress it, and, probably, the truth will be somewhere in between. Therefore, I recommend that the authors also perform LDA calculations (not necessarily for all oxides studied, but maybe for two selected cases) to show whether the ferromagnetism will be found also within the LDA. Also, the value of the lattice constant is important for occurrence of ferromagnetism. Usually, LDA underestimates and GGA overestimates the values of the lattice constant and, therefore, also the existence of ferromagnetism is underestimated in LDA and
overestimated in GGA. A classical example is Fe, see for example the right-hand part of Fig. 1 from the paper
Structure and magnetism of bulk Fe and Cr: from plane waves to LCAO methods By: Soulairol, R.; Fu, Chu-Chun; Barreteau, C. JOURNAL OF PHYSICS-CONDENSED MATTER   Volume: 22   Issue: 29     Article Number: 295502   Published: JUL 28 2010 or an illustrative contour plot of energies of iron as a function of volume and tetragonal deformation both in GGA and LDA in Fig. 2  of the paper
Ab initio calculation of phase boundaries in iron along the bcc-fcc transformation path and magnetism of iron overlayers By: Friak, M; Sob, M; Vitek, V
PHYSICAL REVIEW B   Volume: 63   Issue: 5     Article Number: 052405   Published: FEB 1 2001.
Therefore, as the existence of ferromagnetism is so sensitive to the choice of the approximation for the exchange-correlation potential (LDA vs. GGA)
and to the value of the lattice constant, I strongly recommend that the authors perform such an analysis for at least two selected pyrochlore oxides studied.
An interesting approach overcoming these difficulties in Pd with the help of so-called "volume correction" may be found in the paper
Magnetism and deformation of epitaxial Pd and Rh thin films By: T. Kana et al., PHYSICAL REVIEW B 93, 134422 (2016).
The authors could consider to apply this correction also here
3. end of p. 5:
Here I would expect at least some short Conclusions. Without the
conclusions, no paper is complete! The authors should also present
some outlooks here, rather than at the end of the subsection 3.3.
4. Certainly, the paper
Magnetic pyrochlore oxides Jason S. Gardner, Michel J. P. Gingras, and John E. Greedan Rev. Mod. Phys. 82, 53 – Published 26 January 2010
should be quoted.
Some minor issues:
1. p. 3, line 82: The authors write:
" ... augmented plane wave (FLAPW) scheme [32] and ..."  It would be good to tell the reader also in the text of the article that WIEN2k code has been employed, e.g. in the form " ... augmented plane wave (FLAPW) scheme (WIEN2k code [32]) and ..."
2. p. 3, lines 87 and 88: not "Herman-Feynman force", but "Hellman-Feynman forces" or "Hellman-Feynman forces acting" (not working, line 88).
3. some minor English issues:

not "similar each other", but "similar to each other"
not "This show", but "This shows"
not "this model include", but "this model includes"
not "may not so different", but "may be not so different"
not "Photoemission spectra of this sample does not show",
     but "Photoemission spectra of this sample do not show"
etc. etc.
There are many errors of this type in English grammar and I am not able to indicate all of them. I recommend that a native English speaker reads the
paper before resubmission. To summarize, I recommend a major revision according to the lines given above. After that, the paper may be reconsidered
for publication in Nanomaterials.  

Author Response

See my response in attachment

Reviewer 2 Report

Referee Report

on paper “ Flat-Band in Pyrochlore Oxides: A First-Principles Study “ (nanomaterials-520039) by authors Izumi Hase, Takashi Yanagisawa and Kenji Kawashima submitted to Nanomaterials

This is interesting theoretical paper. It reports the results of first-principles electronic band calculation on obtaining quasi flat-band near the Fermi level for six pyrochlore oxides such as A2B2O7. The band structures of these oxides are well described by non-interacting Mielke model. Spin-polarized calculation shows that the ground state of these compounds is ferromagnetic after appropriate carrier doping, despite the absence of the magnetic element. From my point of view, the most important result is that the ferromagnetic ground state is theoretically established in compounds with the pyrochlore structure that does not contain paramagnetic ions. However, a few points should be improved. I think that this paper can be published only after corresponding additions and corrections :

1.    I understand the choice of the object of study. These are pyrochlores. I fully agree with authors that “ Pyrochlore oxides denoted by A2B2O7 (or A2B2O6O’) have been paid much attention due to their rich variety of physical properties ”. However, there is another class of oxides materials that are also promising for practical use. This is hexaferrites :

(1). M.A. Almessiere, Y. Slimani, H. Güngüne¸ A. Bayka, S.V. Trukhanov, A.V. Trukhanov, Manganese/Yttrium codoped strontium nanohexaferrites: evaluation of magnetic susceptibility and Mössbauer spectra, Nanomat. 9 (2019) 24-18. doi:10.3390/nano9010024.

(2). M.A. Almessiere, Y. Slimani, H.S. El Sayed, A. Baykal, I. Ercan, Microstructural and magnetic investigation of vanadium-substituted Sr-nanohexaferrite, J. Magn. Magn. Mater. 471 (2019) 124-132. https://doi.org/10.1016/j.jmmm.2018.09.054.

This information should be mentioned in 1. Introduction.

2.    The main question is how theoretical calculations took into account the effect of real parameters. First, it is stoichiometry. The deviation of the concentration of the original ions from a given value can lead to a change in the charge state of these ions, which in turn will greatly change the magnetic and electrical parameters. That will seriously affect the practical application of these oxide materials obtained. The most simple is the deviation from oxygen stoichiometry as it is the lightest ion. How was oxygen nonstoichiometry taken into account in theoretical calculations? It is well known that the complex 3d-metal oxides easily allow the oxygen excess and/or deficit :

(3). S.V. Trukhanov, I.O. Troyanchuk, I.M. Fita, H. Szymczak, K. Bärner, Comparative study of the magnetic and electrical properties of Pr1-xBaxMnO3-δ manganites depending on the preparation conditions, J. Magn. Magn. Mater. 237 (2001) 276-282. https://doi.org/10.1016/S0304-8853(01)00477-2.

(4). S.V. Trukhanov, A.V. Trukhanov, A.N. Vasiliev, H. Szymczak, Frustrated exchange interactions formation at low temperatures and high hydrostatic pressures in La0.70Sr0.30MnO2.85, JETP 111 (2010) 209-214. https://doi.org/10.1134/S106377611008008X.

Data on the oxygen stoichiometry of the investigated samples should be discussed in the context of their relationship with magnetic properties.

3.    Oxygen excess and deficit can increase and decrease the oxidation degree of 3d-metalls. The changing of charge state of 3d-metalls as a consequence of changing of oxygen content changes such magnetic parameters as total magnetic moment and Curie point. Moreover, oxygen vacancies effect on exchange interactions. Intensity of exchange interactions decreases with oxygen vacancy concentration increase. In complex oxides there is only indirect exchange. Exchange near the oxygen vacancies is negative according to Goodenough-Kanamori empirical rules. Oxygen vacancies should lead to the formation of a frustration and weak magnetic state such as spin glass :

(5). S.V. Trukhanov, L.S. Lobanovski, M.V. Bushinsky, I.O. Troyanchuk, H. Szymczak, Magnetic phase transitions in the anion-deficient La1-xBaxMnO3-x/2 (0 ≤ x ≤ 0.50) manganites, J. Phys.: Condens. Matter 15 (2003) 1783-1795. https://doi.org/10.1088/0953-8984/15/10/324.

(6). S.V. Trukhanov, A.V. Trukhanov, A.N. Vasiliev, A.M. Balagurov, H. Szymczak, Magnetic state of the structural separated anion-deficient La0.70Sr0.30MnO2.85 manganite. JETP 113 (2011) 819-825. https://doi.org/10.1134/S1063776111130127.

In order for this question to be clarified in the future, it will be necessary to measure the so-called ZFC and FC curves in weak fields up to 100 Oe. This is a reliable way to recognize such a magnetic state as spin glass. The question of separating the state of spin glass and cluster spin glass based on the field exponents remains highly relevant. This information should be discussed in 3.2. Ferromagnetic State and 3.3. Comparison with Experimental Results.

4.    The second question is related to the influence of crystallite size on magnetic and electrical properties. It is well known that for submicron and nanoscale crystallites the magnetic and electrical properties of complex oxides vary significantly with the same chemical composition. Significant contribution is made by dimensional factor and compression of the bulk part of the sample by the surface.

(7). S.V. Trukhanov, A.V. Trukhanov, H. Szymczak, C.E. Botez, A. Adair, Magnetotransport properties and mechanism of the A-site ordering in the Nd-Ba optimal-doped manganites, J. Low Temp. Phys. 149 (2007) 185-199. https://doi.org/10.1007/s10909-007-9507-6.

(8). V.D. Doroshev, V.A. Borodin, V.I. Kamenev, A.S. Mazur, T.N. Tarasenko, A.I. Tovstolytkin, S.V. Trukhanov, Self-doped lanthanum manganites as a phase-separated system: Transformation of magnetic, resonance, and transport properties with doping and hydrostatic compression, J. Appl. Phys. 104 (2008) 093909-9. https://doi.org/10.1063/1.3007993.

How was the influence of the size of a real crystallite taken into account when calculating the magnetic and electrical states? This information should be discussed in 3.2. Ferromagnetic State and 3.3. Comparison with Experimental Results.

5.        The presented 8 papers should be inserted in References.

 The paper should be sent to me for the second analysis after the major revisions.

Author Response

See my response in attachment

Round  2

Reviewer 1 Report

I think that the authors complied with all referees' comments and better explained their views. The paper benefited a lot from the revision and may be accepted in present form.  
Reviewer 2 Report

After my careful evaluation I think that this paper can be accepted in present form.

Nanomaterials EISSN 2079-4991 Published by MDPI AG, Basel, Switzerland RSS E-Mail Table of Contents Alert
Back to Top